# Hepatitis B vaccine uptake and associated factors among adults of Gulu city in Northern Uganda: A community-based cross-sectional study

**Janet Okello Aber**[1]*, **Felix Bongomin**[2], **Stephen Lawoko Opio**[1†], **Emmanuel Ochola**[1,3]

**1** Department of Public Health, Faculty of Medicine, Gulu University, Gulu, Uganda, **2** Department of Medical Microbiology & Immunology, Faculty of Medicine, Gulu University, Gulu, Uganda, **3** Department of HIV, Research and Documentation, St. Mary's Hospital Lacor, Gulu, Uganda

† Deceased.
* aberjanet@gmail.com

## Abstract

**Data Availability Statement:** All relevant data are within the paper.

**Funding:** The Author received no specific funding for this work.

### Background

Hepatitis B virus (HBV) infection is a global public health problem against which vaccination is recommended for all high-risk adults. HBV is highly endemic in Northern Uganda, however, there is a paucity of literature regarding HBV vaccine uptake and associated factors within the community in the region. In this study, we aimed to determine the level of HBV vaccine uptake and associated factors among adults in Gulu city, Uganda.

### Methods

We conducted a community-based, cross-sectional study in Gulu city among eligible adults between March and May 2022. Data on HBV vaccination status and sociodemographic characteristics were collected using an interviewer-administered questionnaire. Full uptake of HBV vaccine was defined as receipt of all 3 recommended doses, and partial uptake for 1 or 2 doses. Multivariable logistic regression analysis was performed using STATA 16.0 to determine factors independently associated with HBV vaccine uptake. P<0.05 was considered statistically significant.

### Results

In total, 360 participants were enrolled, of whom 212 (58.9%) were female, 183 (50.8%) were aged 30 years or younger, and 143 (39.7%) had attained tertiary education. Overall, 96 (26.7%) participants had full uptake of HBV vaccine and 73 (20.3%) had partial uptake. Factors that were statistically significantly associated with full uptake of HBV vaccine were good knowledge regarding HBV transmission (aOR = 1.9, 95% Confidence Interval (CI) = 1.03–3.46, p = 0.040) and receiving health education on HBV vaccination (aOR = 4.4, 95% CI = 2.3–8.4, p<0.001).

**Competing interests:** the Authors have declared that no competing interests exist.

**Abbreviations:** HBV, Hepatitis B Viral Infection; CDC, Centers for Disease Control and prevention; MOH, Ministry of Health; UPHIA, Uganda Population based HIV Impact Assessment; WHO, World Health Organization.

## Conclusions

There is a low uptake of HBV vaccine in Gulu city, Uganda. The Uganda Ministry of Health should correct misconceptions, create awareness of the severity of HBV infection through health education regarding HBV infection within the community in Gulu City; and set mechanisms to follow-up clients due for next HBV vaccination.

## Introduction

Hepatitis B virus (HBV) is an hepatotropic DNA virus with potential for oncogenesis [1]. Globally, infection due to HBV is the most common cause of viral hepatitis, chronic liver disease and hepatocellular carcinoma [2]. At the end of 2019, an estimated 297 million people were living with HBV resulting to over 820,000 HBV-related deaths globally [3]. The global prevalence of HBV within the general population is 3.8%, with an estimated 1.5 million annual incident cases [3]. The World Health Organization (WHO) African region has the highest prevalence of HBV infection at 7.5%, which translates to over 82.3 million cases [3].

HBV burden in Uganda was reported at 4.1% among adults, prevalence being highest in the mid North region (4.6%), North-Eastern region (4.4%), and the West-Nile (3.8%) [4]. Within the general population of Gulu, HBV prevalence was estimated at 17.6% with a lifetime exposure of 72.4% [5]. The prevalence was at 11.8% among pregnant women in Gulu [6]. However, of the few studies that have attempted to investigate the uptake of HBV vaccine in Uganda, a prevalence of 44.3% and 57.8% were reported among medical students at Makerere University and Health providers in Wakiso, respectively [7, 8].

HBV chronic infection has been the most important outcome of public health interest, but there has been recent evolutions in the mortality burden of HBV outcomes which demands for a reform in the focus of prevention [9]. Acute HBV infection may result into fulminant hepatitis, which can progress to acute liver failure (ALF) leading to death if liver transplantation is not urgently done [10]. It is responsible for 17% of all deaths due to HBV while liver cancer and cirrhosis caused 43% and 40% respectively [11]. This mortality rate for acute HBV demand that HBV prevention targets all individuals at high risk especially adults living in areas with high endemicity. Due to prenatal infection to infants who consequently become chronic carriers, preventing HBV infection among the general population is of public health importance.

HBV vaccination is the single most effective method of preventing HBV infection [12]. On top of the childhood pentavalent vaccination inclusive of Hepatitis B vaccine, the WHO recommends HBV vaccination for all at risk populations [13, 14].

However, among the population born before HBV vaccine was introduced, only a few people have taken up vaccination as a preventive measure. Numerous studies done in various parts of Africa on acceptance and completion of HBV vaccination revealed poor outcomes generally especially in developing countries [15, 16]. Low uptake of HBV vaccination was also reported among Federal Road Safety Corps in Nigeria at 30.5% despite the high initiation rate of 60.9%. The participants who had the feeling that their occupation could expose them to HBV were 3 times more likely to receive vaccination. Inadequate knowledge regarding the number of doses of the vaccine that provides lifetime protection was significantly associated with low completion rate [9].

Similarly, in Nigeria there was low uptake of HBV vaccination (27%) among university students even though the university took serious efforts to educate the students about hepatitis B and including the vaccination cost in their fees. Despite the students' awareness of HBV

infection, their knowledge didn't motivate their actions or practice since 44% did not receive any prior vaccination. The students' perceived susceptibility and the need to be protected from HBV, were factors that positively influenced completion of the dosage of the HBV vaccination. Factors that limited the students from completing the immunization dosages were inadequate knowledge concerning the required doses for full immunity, forgetfulness, the feeling that the vaccine was not necessary, and fear of the possible side effects of the vaccine [17]. In Wakiso district in Uganda, more than half (57.8%) of the health workers were fully vaccinated. Being knowledgeable about the vaccine safety and effectiveness and working in a private facility influenced full vaccination [8]. There is currently no published research on the uptake of HBV vaccine and associated factors among the general adult population in Uganda.

According to the Uganda Guidelines for Prevention, Testing, Care and Treatment for Hepatitis B, all newborn infants should receive the birth dose and the pentavalent vaccine given at 6,10 and 14 weeks. Adult vaccines should be given at 0,1 and 6 months to all persons at high risk of HBV including all health workers, pregnant women, persons living with HIV or other sexually transmitted infections, household contacts of an infected person, sexual contacts of people living with chronic HBV, armed forces, prisoners, people with sickle cell disease or other patients who frequently receive blood or blood products, sex workers, people with multiple sexual partners, injection drug users and men who have sex with men [18].

Uganda being one of the countries that adopted the WHO prevention strategies against HBV introduced HBV vaccine in the immunization schedule in 2002 for immunization of children 6 weeks to 5 years old [18]. To effectively prevent HBV new infections in Uganda, multiple approaches need to be considered. Therefore, vaccination of adults in highly endemic areas should complement prevention of perinatal transmission. Recently the MOH, Uganda set up a region based universal HBV vaccination of adults in Uganda starting with Northern Uganda since it is the region with the highest endemicity completely free of charge but the uptake was still low [19].

Many people in northern Uganda are living unaware of their infection status and hence continue transmitting the virus to others unknowingly. Even though the vaccine had been made available by the MOH, only 33% of the people completed the three recommended doses for full immunity [19]. Since this region has the highest burden of HBV infection, normally we would expect the public to fully embrace the vaccination programme. Gaps in knowledge, practice and uptake of HBV vaccine still exist, which have led to vaccine hesitancy. This means the population susceptible to the infection remains high. The reasons for the low HBV vaccination uptake are unclear.

Increasing HBV vaccination uptake would possibly result in decreasing HBV infection transmission in Uganda. Therefore, studying HBV vaccine uptake and associated factors in Gulu City was important to understand the challenges hindering people from embracing vaccination and completion of the 3 doses of the vaccine. The study has improved the understanding on key determinants of hepatitis B vaccination uptake and will inform policy so that possible interventions to improve uptake can be put in place. The result of this study will be used by programme planners and implementers to raise awareness regarding HBV vaccination and completion for the prevention of HBV infection. It was thus necessary to conduct this study to determine HBV vaccine uptake and associated factors among adults of Gulu City.

## Materials and methods

### Study design

A community-based, descriptive, observational study using a cross-sectional design was conducted between March and May 2022 in Gulu City, Uganda.

## Study setting

Gulu, elevated to city status in 2021, is in Northern Uganda, being bordered by Lamwo district to the North, Pader and Omoro districts to the East, Oyam district to the South, Nwoya district to the Southwest, and Amuru district to the West. The city consists of 32 parishes and 128 villages with a total population of 271,049 people. The study population included all adults in Gulu City who were available during the data collection period and satisfied the selection criteria.

## Sample size determination

The modified Kish Leslie formula was used to determine the sample size (n = $Z^2$ * P(1-P)/$e^2$). The confidence level was set at 95%, and a margin of error (e) of 5%. The proportion (P) of vaccinated people in Northern Uganda was 33% [19]. The sample size was therefore estimated at 340 participants. To make up for anticipated incomplete information, 20 participants were added, thus making a total of 360 participants.

## Sampling method

The samples were obtained by random sampling (using table of random numbers) of 16/32 (50%) parishes from which participants were proportionately sampled. Households were systematically randomly sampled, taking the K[th] household unit until the sample size was met. Only one adult per household was randomly sampled using simple random sampling among those who were present at home at the time of the data collection.

## Data collection

Data was collected using pretested structured questionnaires administered by trained research assistants who were fluent in both English and *Luo -leb Acoli* (native language of the Luo people in Gulu city) languages. Participants were recruited from 29[th] March to 13[th] May 2022. Data was collected on sociodemographic characteristics, HBV knowledge, health systems factors and HBV vaccination.

## Operational definitions

1. Full uptake of HBV vaccine: A participant who received all the 3 recommended doses of the vaccine according to the WHO/MoH guidelines.

2. Partial uptake of HBV vaccine: A participant who received 1 or 2 doses of the 3 recommended doses of the vaccine according to the WHO/MoH guidelines.

3. Non- uptake: A participant who received none of the 3 recommended doses of the vaccine according to the WHO/MoH guidelines.

## Data analysis

Data was analyzed using STATA version 16.0. Continuous variables (age, number of sexual partners and household income) were described using median and range while categorical variables (HBV vaccine uptake and associate factors) were described using frequencies and percentages. The results were displayed using tables and graphs.

At bivariate analysis, association between HBV vaccine uptake and the independent factors was assessed using Chi-square or Fischer's exact tests for categorical variable. Variables with p<0.2 at bivariate analysis or biologically plausible to be associated with HBV vaccine uptake

were forward to multivariable analysis. A stepwise, binary logistic regression model was constructed to determine factors that were independently associated with HBV vaccine uptake after accounting for all important confounders. Factors with p< 0.05 were considered statistically, significantly, independently associated with HBV vaccine uptake. Results were presented as adjusted odds ratio (aOR) and 95% confidence interval (95%CI).

## Ethical considerations

Ethical clearance was sought from Gulu University Research and Ethics Committee (GUREC) which is accredited by the Uganda National Council for Science and Technology (approval number GUREC-2021-179. All the methods were performed in accordance with the ethical guidelines and regulations outlined in the declaration of Helsinki. All the respondents gave voluntary, written informed consent following adequate information prior to the study participation. Confidentiality was ensured throughout the study and privacy provided during the interview.

## Results

### Baseline characteristics of the study participants

A total of 360 eligible participants constituted the study population. Female gender comprised 212 (58.9%), median age was 29 (range: 18–76) years. Overall, 203 (56.4%) participants were married and 291 (80.8%) were Acholi by tribe. Most participants 143(39.7%) had attained tertiary education and 184 (51.1%) were unemployed, Table 1.

### Hepatitis B vaccination status

Overall, 96 (26.7%) participants received full and 73 (20.3%) partial (one or two) HBV vaccine doses, Fig 1. The key reasons for non-completion of vaccination schedules included unavailability of vaccines (28%), forgetting the next appointment date (22.1%), and being too busy to go for vaccination (8.8%), cost of vaccine (7.4%) and was told that two doses were enough (7.4%).

### Factors associated with HBV vaccine uptake

Table 2 summarizes the results from the multivariate analysis of Hepatitis B vaccine uptake and associated factors. Knowledge of HBV transmission routes (p = 0.040) and health education (p<0.001) were significantly associated with HBV vaccine uptake. Participants who were knowledgeable regarding HBV transmission were 2 times more likely to be vaccinated (aOR = 1.888, 95%CI = 1.031–3.459) than those who were not knowledgeable about HBV transmission routes. Participants who had ever been health educated by health workers regarding HBV vaccine were 4 times more likely to vaccinate (aOR = 4.387, 95% CI = 2.297–8.379) than those who had never received any health education regarding HBV vaccine.

## Discussion

In this study, we found a low uptake of HBV vaccination among adults of Gulu City in Northern Uganda, with only about one in four (26.7%) receiving 3 doses, 12.2% two doses, and 8.1% 1 dose. This low vaccination uptake is a serious public health challenge for a region with high HBV prevalence like Gulu city and thus increased transmission risk [4, 5]. This calls for urgent interventions by policy makers to increase HBV vaccine uptake. Our findings are consistent with 27% uptake among University students in Nigeria [17], 24.6% among adults ≥19 years in the United States [20] and 22.1% among health care workers in Juba [21].

**Table 1. Socio-demographic and other characteristics of the participants.**

| Variable | Categories | Frequency | Percentage |
|---|---|---|---|
| **Age** | <30 | 183 | 50.8 |
| | ≥30 | 177 | 49.2 |
| **Sex** | Male | 148 | 41.1 |
| | Female | 212 | 58.9 |
| **Marital status** | Single | 115 | 31.9 |
| | Married | 203 | 56.4 |
| | Divorced | 32 | 8.9 |
| | Widowed | 10 | 2.8 |
| **Tribe** | Acholi | 291 | 80.8 |
| | Langi | 32 | 8.9 |
| | Others* | 37 | 10.3 |
| **Education level** | Informal | 19 | 5.3 |
| | Primary | 76 | 21.1 |
| | Secondary | 122 | 33.9 |
| | Tertiary | 143 | 39.7 |
| **Employment status** | Employed | 176 | 48.9 |
| | Unemployed | 184 | 51.1 |
| **Level of income** | 0–200,000 | 236 | 65.6 |
| | 201,000–500,000 | 69 | 19.2 |
| | 501,000–1,000,000 | 43 | 11.9 |
| | 1,000,001 + | 12 | 3.3 |
| **Blood transfusion** | No | 297 | 82.5 |
| | Yes | 63 | 17.5 |
| **Sexual partners** | 0–1 | 151 | 41.9 |
| | 2+ | 209 | 58.1 |
| **Unprotected sex** | No | 71 | 19.7 |
| | Yes | 189 | 80.3 |
| **Contact with body fluid** | No | 177 | 49.2 |
| | Yes | 183 | 50.8 |
| **HIV status** | Negative | 341 | 94.7 |
| | Positive | 19 | 5.3 |
| **Ever had a sexually transmitted infection** | No | 309 | 85.8 |
| | Yes | 51 | 14.2 |

Others* Alur 4, Musoga 1, Karamojong 4, Lugbara 3, Madi 6, Muganda 4, Mukiga 1, Munyankole 2, Kuku 1, Itesot 6, Mufumbira 1, Munyole 1, Munyoro 1, Sabin 1. 1 United States Dollar = 3,750 Ugandan shillings

Our study revealed that not all individuals who start the vaccine go on to finish, which has implications for follow-up of those who start the vaccination. We found that, about one in five, (20.3%) was partially vaccinated and the major reasons for incomplete vaccination included unavailability of vaccines, forgetting the next appointment date, being too busy to go for vaccination, being told that two doses were enough and the vaccine being expensive. Numerous epidemiological studies elsewhere also acknowledged incomplete vaccination [17, 22, 23].

Household members may be a good reminder system for the next vaccine dose due date as already seen for HIV drug refills [24]. The health workforce could also use phone calls or local leaders to remind their communities whenever the date for the next vaccine doses elapsed.

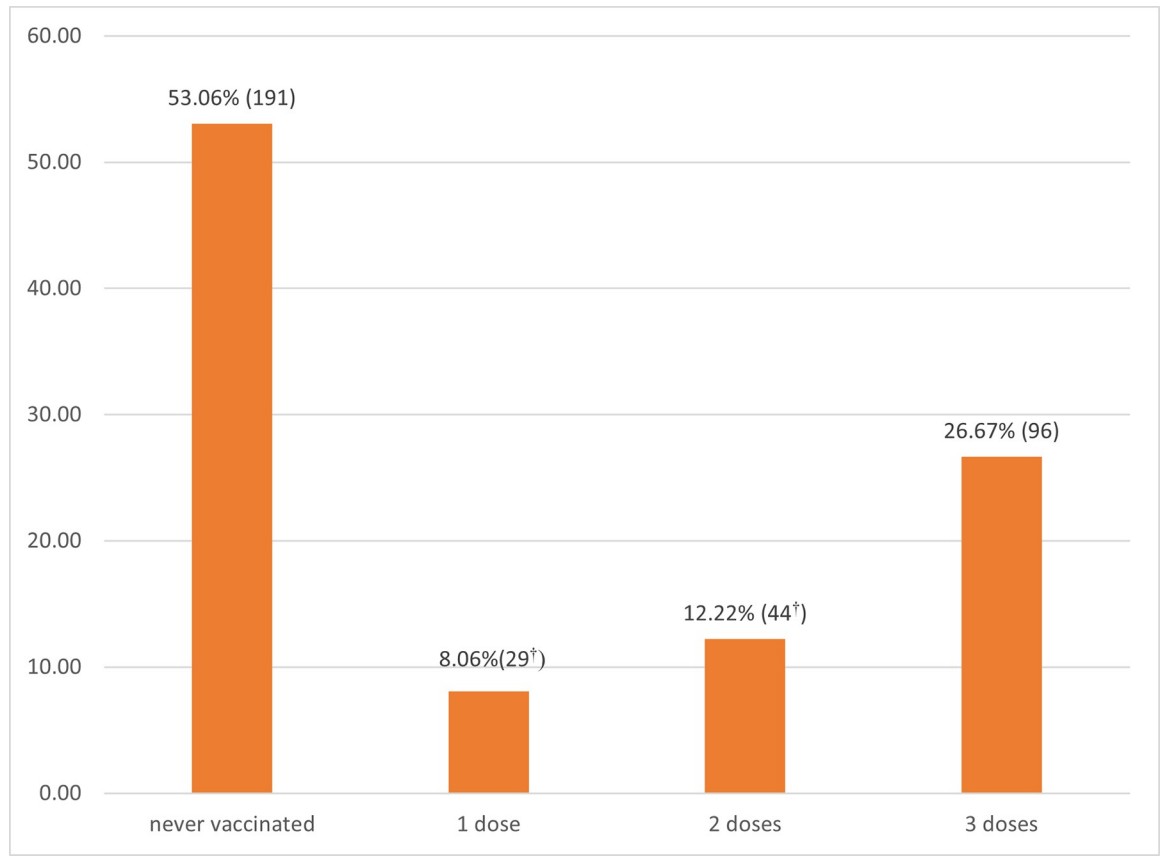

**Fig 1. Hepatitis B vaccination status of the study participants (n = 360).** † The top five reasons for non-completion included unavailability of vaccines (28%), forgetting the next appointment date (22.1%), and being too busy to go for vaccination (8.8%), cost of vaccine (7.4%) and was told that two doses were enough (7.4%).

The factors that were found to be significantly associated with uptake in this study were access to health education and knowledge regarding HBV transmission. Among the participants who had received health education, about two of every five (44.1%) were vaccinated. The odds of vaccination was four times higher (95% CI = 2.297–8.379) than those who didn't receive health education about HBV vaccination. This suggests that it is crucial for the community, families and individuals to understand the importance and benefits of taking actions to prevent HBV through vaccination. Other studies in Malaysia, Nigeria and Vietnam also found that knowledge of HBV infection and the vaccine influenced vaccination uptake [9, 25, 26]. Our focus should therefore be geared towards activities that will encourage individuals to take up vaccination. There is need for health education and mass media campaigns.

Knowledge regarding HBV transmission was also significantly associated with HBV vaccine uptake. This variable was assessed using two categories. Participants who stated at least one correct method of transmission were categorized as "knowledgeable," while those who didn't know any method of transmission and those who wrongly stated all the transmission methods were categorized as "not knowledgeable." This variable increased the probability of vaccination by about 2 times.

Only 42.2% of the participants were knowledgeable about HBV transmission routes. 28.3% of the participants wrongly stated HBV transmission routes, including sweat, saliva, contaminated food and water, airborne, sneezing, sharing bathroom, dirtiness, body contact, sharing

**Table 2. Multivariable analysis of HBV vaccine uptake and associated factors.**

| Variable | N = 360 Frequency (%) | Univariable | | | Multivariable | | |
|---|---|---|---|---|---|---|---|
| | | Crude OR | 95%CI | P-value | AOR | 95%CI | P-value |
| **Individual factors** | | | | | | | |
| **Vaccination is the best way to prevent HBV** | | | | | | | |
| No | 53(14.7) | | | | | | |
| Yes | 307(85.3) | 7.243 | 2.203–23.813 | **0.001** | 2.90 | 0.754–11.122 | 0.121 |
| **Necessity to get vaccinated** | | | | | | | |
| No | 45(12.5) | | | | | | |
| Yes | 315(87.5) | 9.145 | 2.171–38.527 | **0.003** | 4.723 | 0.937–23.817 | 0.060 |
| **Knowledge regarding HBV transmission** | | | | | | | |
| Not knowledgeable | 208(57.8) | | | | | | |
| Knowledgeable | 152(42.2) | 2.310 | 1.435–3.716 | **0.001** | 1.888 | 1.031–3.459 | **0.040** |
| **Knowledge about effects of HBV** | | | | | | | |
| Not knowledgeable | 43(11.9) | | | | | | |
| Knowledgeable | 307(88.1) | 5.536 | 1.671–18.341 | **0.005** | 3.448 | 0.886–13.421 | 0.074 |
| **HBV affects liver only** | | | | | | | |
| No | 121(33.6) | | | | | | |
| Yes | 239(66.4) | 1.743 | 1.030–2.947 | **0.038** | 1.297 | 0.652–2.578 | 0.458 |
| **Safety of HBV vaccine** | | | | | | | |
| Not safe | 111(30.8) | | | | | | |
| Safe | 259(69.2) | 3.402 | 1.831–6.321 | **0.000** | 1.666 | 0.738–3.758 | 0.219 |
| **HBV vaccine efficacy** | | | | | | | |
| No | 101(28.1) | | | | | | |
| Yes | 259(71.9) | 1.987 | 1.119–3.530 | **0.019** | 0.850 | 0.399–1.809 | 0.673 |
| **Health system factors** | | | | | | | |
| **Ability to vaccinate from private hospital** | | | | | | | |
| No | 176(48.9) | | | | | | |
| Yes | 184(51.1) | 1.998 | 1.235–3.231 | **0.005** | 1.582 | 0.866–2.888 | 0.135 |
| **Ever been health educated** | | | | | | | |
| No | 183(50.8) | | | | | | |
| Yes | 177(49.2) | 7.222 | 4.085–2.769 | **<0.001** | 4.387 | 2.297–8.379 | **<0.001** |
| **Ever been mobilized** | | | | | | | |
| No | 90(25) | | | | | | |
| Yes | 270(75) | 2.143 | 1.162–3.953 | **0.015** | 1.743 | 0.809–3.754 | 0.156 |

plates and utensils, smoking, drinking alcohol, sitting together, sharing clothes and greetings. This finding is similar to a study by Hislop and colleagues [27] where only a few participants recognized that it was not a food borne disease. This limited knowledge is also in line with previous studies where most participants were unaware that HBV is transmitted through unprotected sex and from mother to child [28–30]. Could it be that, if people knew that they could transmit the virus to their children or partners, they would take it more seriously and vaccinate? There is need to provide correct information to the population to increase uptake of HBV vaccination.

In our sub analysis, we explored major reasons for non-vaccination. Among the participants who were not vaccinated at all, lack of trust for the vaccine was the most stated reason by 3 of every 10 participants, followed by declining to vaccinate (20.9%) thus differing from a study by Park and colleagues [31] which showed that the most cited reason for non-

vaccination was lack of knowledge regarding the necessity for HBV vaccination and that of Ferreira and colleagues [32], who found that lack of information was the main reason for not vaccinating. However, our finding is lower than the findings from France where 39% didn't trust the vaccine [33] but similar to 27% in Berlin dental personnel who stated that the vaccine was not safe [34]. About 19.4% of the participants in this study were not aware of the mass vaccination program and 10.5% were too busy which prevented them from taking up vaccination. HBV vaccine hesitancy could also be due to the widespread sale of falsified HBV vaccines on the Ugandan market in 2018. There seems to be some mistrust still existing within the communities [35]. There is need to build confidence and trust in the population to increase HBV vaccine uptake in Gulu city.

## Strengths and limitations

The study has several limitations. Vaccination status was based on self-report by the respondents and not by reviewing their vaccination cards. There is a possibility that the study suffered recall bias since they may have not remembered accurately their vaccination history thus introducing information bias, which could be non-differential. This was minimized by giving the interviewee sufficient time to allow them adequate recall. Being a cross sectional study, suitability for demonstrating temporal relationships between explanatory and outcome variable is limited. However, it showed independent associations useful in understanding the factors associated with vaccination uptake among the study population which will inform relevant public health interventions. However, pretesting of questionnaires provided for clarity of the questions and increased the response rate. Confounding, one of the most important problems in observational studies was minimized through multivariate logistic regression analysis. The ample sample size used made it representative of the population and the response rate of 100% minimized selection bias thus enhancing generalizability of the research findings. Another limitation is the time of conducting the survey as this could affect the category of participants available and present in the various households. This was minimized by collecting data as early as 8:00 am to 6:00 pm from Monday to Saturday.

## Conclusions

The uptake of HBV vaccination among adults in Gulu city is generally low. Only approximately one in four (26.7%) of the respondents were fully vaccinated with three doses of the vaccine. However, an additional 20.3% percent started, but were only partially vaccinated. The factors found to be independently associated with HBV vaccine uptake were health education and knowledge regarding HBV transmission.

The findings have implications on the need for strategic and factual health education regarding HBV infection, prevention and treatment for the people of Gulu city to increase awareness and knowledge of the seriousness of the disease in order to reduce the burden of HBV infection. Health workers need to initiate a client follow up system for subsequent vaccination doses. The low vaccination uptake, coupled with the high prevalence of HBV in northern Uganda require strategies to increase HBV vaccine uptake among adults.

## Acknowledgments

We are grateful to all the research assistants and the study participants.

## Author Contributions

**Conceptualization:** Janet Okello Aber, Emmanuel Ochola.

**Data curation:** Janet Okello Aber.

**Formal analysis:** Janet Okello Aber, Emmanuel Ochola.

**Funding acquisition:** Janet Okello Aber.

**Investigation:** Janet Okello Aber.

**Methodology:** Janet Okello Aber, Felix Bongomin, Stephen Lawoko Opio, Emmanuel Ochola.

**Project administration:** Janet Okello Aber.

**Resources:** Janet Okello Aber.

**Software:** Janet Okello Aber.

**Supervision:** Felix Bongomin, Stephen Lawoko Opio, Emmanuel Ochola.

**Validation:** Janet Okello Aber.

**Visualization:** Janet Okello Aber.

**Writing – original draft:** Janet Okello Aber, Felix Bongomin, Emmanuel Ochola.

**Writing – review & editing:** Janet Okello Aber, Felix Bongomin, Emmanuel Ochola.

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
