## [Decision Letter · Decision Letter 0]

6 Nov 2023

PONE-D-23-29648Hepatitis B vaccine uptake and associated factors among adults of Northern Uganda: A community-based cross-sectional study.PLOS ONE

Dear Dr. Aber,

Thank you for submitting your manuscript to PLOS ONE. After careful consideration, we feel that it has merit but does not fully meet PLOS ONE’s publication criteria as it currently stands. Therefore, we invite you to submit a revised version of the manuscript that addresses the points raised during the review process.

We look forward to receiving your revised manuscript.

Kind regards,

Jason T. Blackard, PhD

Academic Editor

PLOS ONE

Journal Requirements:

Whilst you may use any professional scientific editing service of your choice, PLOS has partnered with both American Journal Experts (AJE) and Editage to provide discounted services to PLOS authors. Both organizations have experience helping authors meet PLOS guidelines and can provide language editing, translation, manuscript formatting, and figure formatting to ensure your manuscript meets our submission guidelines. To take advantage of our partnership with AJE, visit the AJE website (http://aje.com/go/plos) for a 15% discount off AJE services. To take advantage of our partnership with Editage, visit the Editage website (www.editage.com) and enter referral code PLOSEDIT for a 15% discount off Editage services. If the PLOS editorial team finds any language issues in text that either AJE or Editage has edited, the service provider will re-edit the text for free.

5. Please amend either the title on the online submission form (via Edit Submission) or the title in the manuscript so that they are identical.

Additional Editor Comments:

This is a cross-sectional, community-based study of HBV vaccination in Northern Uganda.

Overall, the results of this study are not surprising and the writing would benefit from review by a native English speaker and/or a professional editing service.

Were number of HBV vaccine doses received self-reported by the study participants or extracted from medical records?

Were HBV antibodies measured / quantified to know if the vaccination series worked effectively?

It is not clear how knowledge of HBV transmission routes and/or health education were provided in the first place.  Were these part of the study design or provided in some other setting?

Reviewers' comments:

Reviewer's Responses to Questions

**Comments to the Author**

1. Is the manuscript technically sound, and do the data support the conclusions?

Reviewer #1: Partly

Reviewer #2: Yes

2. Has the statistical analysis been performed appropriately and rigorously? 

Reviewer #1: Yes

Reviewer #2: Yes

3. Have the authors made all data underlying the findings in their manuscript fully available?

Reviewer #1: Yes

Reviewer #2: Yes

4. Is the manuscript presented in an intelligible fashion and written in standard English?

Reviewer #1: Yes

Reviewer #2: Yes

5. Review Comments to the Author

Reviewer #1: Title: This is misleading as Gulu city is not equivalent to Northern Uganda. it should read" Gulu City in Northern Uganda"

Methods: There should be more description or details of the proportional sampling of the parishes and systematic random sampling of households. What time of the day were questionnaires administered as the time of day can also determine the types of respondents.

Results:

figure 2-No need for 2 bars ie freq/percentage. You can use percentages and put the figure in brackets on top of the bar.

figure 3:No need for a bar chart. the data can be presented in a table and it will be clear.

table 2: Multivariable analysis-The table can be improved by including only variables that had significant p values in the univariate analysis

No need to write AOR in full in results. Use of times ie 4 times makes sentence clearer in this case as opposed to 4 folds

Discussion: There is quite an amount of repetition of results in this section. This should be addressed and reduced. The discussion could also be beefed up.

Limitations: Another limitation of the study is the time of conducting the survey as this could affect the category of respondents available and present in the various households.

General:There are grammatical errors which should be corrected. Hepatitis B, the B should be in capitals

Reviewer #2: The manuscript is a basic public health study. The study should have also explain Uganda's policy on HBV vaccination policy on infants and adults within the background section. This should then be linked with adult population factors contributing to failure to get vaccinated. Table 1: how many of these study participants were healthcare workers? Table 1: Level of income, what currency is. Is there any difference between children population and adult regarding vaccination status? While this study may give an impression of significance, unfortunately its more applicable to Uganda policy makers, and more relevant there. It does not really add new info to current knowledge of HBV vaccination

6. PLOS authors have the option to publish the peer review history of their article (what does this mean?). If published, this will include your full peer review and any attached files.

Reviewer #1: No

Reviewer #2: **Yes: **A Lukhwareni

---

## [Author Response · Author response to Decision Letter 0]

13 Dec 2023

COMMENTS TO REVIEWERS

Reviewer #1: Title: This is misleading as Gulu city is not equivalent to Northern Uganda. it should read" Gulu City in Northern Uganda"

Methods: There should be more description or details of the proportional sampling of the parishes and systematic random sampling of households. What time of the day were questionnaires administered as the time of day can also determine the types of respondents.

Authors’ response: We agree with the reviewer. The title was amended reading ‘Hepatitis B vaccine uptake and associated factors among adults of Gulu City in Northern Uganda. A community based cross sectional study.

Methods: A total of 16 of the 32 (50%) parishes in Gulu City were randomly sampled using number tables. Stratified random sampling proportionate to size was done to determine the number of participants from each parish.

Random samples were taken from each parish in proportion to the population in that parish. We used the proportionate stratified random sampling formula: nh = (Nh / N) *n where nh =sample size for each Parish, Nh =population size in each parish, N =size of the entire population in Gulu City, n = size of the entire sample.

Systematic random sampling: we calculated the sampling interval by dividing the entire population size of that parish by the desired sample size for that parish. Sampling of the households began from anywhere selected at random.

Questionnaires were administered from 8:00am to 6:00pm from Monday to Saturday.

Results:

figure 2-No need for 2 bars i.e., freq/percentage. You can use percentages and put the figure in brackets on top of the bar.

Authors’ response: I have removed 1bar and for data label, I have used percentages and figures in brackets. 

Figure 3: No need for a bar chart. the data can be presented in a table and it will be clear.

Authors’ response: Thank you for the suggestion, we have removed the bar chart and incorporated it in figure two. Its contents have been put as a caption under figure two.

Table 2: Multivariable analysis-The table can be improved by including only variables that had significant p values in the univariate analysis

No need to write AOR in full in results. Use of times ie 4 times makes sentence clearer in this case as opposed to 4 folds

Authors’ response: table 2 was modified. Only variables that had significant p values were included.

The words adjusted odds ratio were removed.

The use of 4 folds were eliminated and replaced by 4 times.

Discussion: There is quite an amount of repetition of results in this section. This should be addressed and reduced. The discussion could also be beefed up.

Authors’ response: We agree with the reviewer. The discussion section was edited.

Limitations: Another limitation of the study is the time of conducting the survey as this could affect the category of respondents available and present in the various households.

Authors’ response: Thank you, this was included under limitations.

General: There are grammatical errors which should be corrected. Hepatitis B, the B should be in capitals

Authors’ response: Grammatical errors were corrected. The B in Hepatitis B has been capitalised throughout the manuscript.

Reviewer #2: The manuscript is a basic public health study. The study should have also explained Uganda's policy on HBV vaccination policy on infants and adults within the background section. This should then be linked with adult population factors contributing to failure to get vaccinated.

Authors’ response: Thank you, we agree with the reviewer and have added a substantial amount of information in the introduction. Uganda policy on hepatitis B prevention, care and treatment has been included in the background section. A few factors contributing to low vaccine uptake have also been included in the background.

Table 1: how many of these study participants were healthcare workers? Table 1: Level of income, what currency is. Is there any difference between children population and adult regarding vaccination status? While this study may give an impression of significance, unfortunately its more applicable to Uganda policy makers, and more relevant there. It does not really add new info to current knowledge of HBV vaccination

Authors’ response: Good insight. We are not sure how many of the study participants were healthcare workers. The study was done in households within the community and definitely some of the participants might have been healthcare workers.

Table 1: under level of income, the currency used was Uganda shillings. I have indicated it in table 1 and a conversion of USD to UGX has been indicated.

Regarding HBV vaccination status, in Uganda, children born starting from 2002 onwards received the pentavalent vaccine DPT which has a combination of DPT, Hep B, HCV. Individuals born before 2002 were not vaccinated against HBV.

---

## [Decision Letter · Decision Letter 1]

10 Jan 2024

Hepatitis B vaccine uptake and associated factors among adults of Gulu City in Northern Uganda: A community-based cross-sectional study.

PONE-D-23-29648R1

Dear Dr. Aber,

We’re pleased to inform you that your manuscript has been judged scientifically suitable for publication and will be formally accepted for publication once it meets all outstanding technical requirements.

Kind regards,

Jason T. Blackard, PhD

Academic Editor

PLOS ONE

Additional Editor Comments (optional):

None

Reviewers' comments:

Reviewer's Responses to Questions

**Comments to the Author**

1. If the authors have adequately addressed your comments raised in a previous round of review and you feel that this manuscript is now acceptable for publication, you may indicate that here to bypass the “Comments to the Author” section, enter your conflict of interest statement in the “Confidential to Editor” section, and submit your "Accept" recommendation.

Reviewer #1: All comments have been addressed

2. Is the manuscript technically sound, and do the data support the conclusions?

Reviewer #1: (No Response)

3. Has the statistical analysis been performed appropriately and rigorously? 

Reviewer #1: (No Response)

4. Have the authors made all data underlying the findings in their manuscript fully available?

Reviewer #1: (No Response)

5. Is the manuscript presented in an intelligible fashion and written in standard English?

Reviewer #1: (No Response)

6. Review Comments to the Author

Reviewer #1: (No Response)

7. PLOS authors have the option to publish the peer review history of their article (what does this mean?). If published, this will include your full peer review and any attached files.

Reviewer #1: No

---

## [Editor Report · Acceptance letter]

21 Feb 2024

PONE-D-23-29648R1 

PLOS ONE

Dear Dr. Aber, 

I'm pleased to inform you that your manuscript has been deemed suitable for publication in PLOS ONE. Congratulations! Your manuscript is now being handed over to our production team.

Kind regards, 

on behalf of

Dr. Jason T. Blackard 

Academic Editor

PLOS ONE